

# The smoother the better?
# A comparison of six post-processing methods to improve short-term offshore wind power forecasts in the Baltic Sea

Christoffer Hallgren[1], Stefan Ivanell[1], Heiner Körnich[2], Ville Vakkari[3,4], and Erik Sahlée[1]

[1]Department of Earth Sciences, Uppsala University, Uppsala, Sweden
[2]Swedish Meteorological and Hydrological Institute, Norrköping, Sweden
[3]Finnish Meteorological Institute, Helsinki, Finland
[4]Atmospheric Chemistry Research Group, Chemical Resource Beneficiation, North-West University, Potchefstroom, South Africa

**Correspondence:** Christoffer Hallgren (christoffer.hallgren@geo.uu.se)

**Abstract.** With a rapidly increasing capacity of electricity generation from wind power, the demand for accurate power production forecasts is growing. To date, most wind power installations have been onshore and thus most studies on production forecasts have focused on onshore conditions. However, as offshore wind power is becoming increasingly popular it is also important to assess forecast quality in offshore locations. In this study, forecasts from the high-resolution numerical weather prediction model AROME was used to analyze power production forecast performance for an offshore site in the Baltic Sea. To improve the AROME forecasts, six post-processing methods were investigated and their individual performance analyzed in general as well as for different wind speed ranges, boundary layer stratifications, synoptic situations and in low-level jet conditions. In general, AROME performed well in forecasting the power production, but applying smoothing or using a random forest algorithm increased forecast skill. Smoothing the forecast improved the performance at all wind speeds, all stratifications and for all synoptic weather classes, the random forest method increased the forecast skill during low-level jets. To achieve the best performance, we recommend to select which method to use based on the forecasted weather conditions. Combining forecasts from neighbouring grid points, combining the recent forecast with the forecast from yesterday or applying linear regression to correct the forecast based on earlier performance were not fruitful methods to increase the overall forecast quality.

## 1 Introduction

With a growing concern about a future climate crisis and a continuously increasing demand for electrical power, a greater penetration of renewable energy sources in the power supply system becomes crucial to meet the climate goals (e.g., Sims, 2004; Quaschning, 2019). In the last 20 years, more and more attention has been directed to wind power and as new technical inventions have enabled construction of larger and more efficient turbines this technique now also accounts for a bigger share of the total power production (Lee and Zhao, 2021).

As wind power production is highly dependent on the weather, a deep understanding of the climate for a site is needed when assessing the optimal location for a new wind farm, taking both the average and extreme conditions into account. Once the



farm is in operational use, the meteorological focus shifts from site climatology to weather forecasting, to be able to predict the instantaneous power production. Accurate forecasts, especially for the short perspective (minutes to hours) but also for longer time scales (weeks to seasons), are requested by the grid operators, power production companies and traders on the electricity

market to balance the power in the grid, to plan ahead and to maximize the revenue (e.g., Foley et al., 2012; Heppelmann et al., 2017; Lledó et al., 2019).

When it comes to planning for new wind power farms, offshore sites have recently gained more attention (Esteban et al., 2011; Díaz and Soares, 2020). The main reason for this is simple: as the wind speed is generally higher over water than over land, constructing a new wind farm offshore allows for higher production. However, similar as when planning for a wind farm

onshore, there are many aspects that must be considered before a new farm can become a reality, such as noise and visual disturbances (e.g., Bishop and Miller, 2007), military restricted areas, natural resources and animal life (e.g., Leung and Yang, 2012), topography/bathymetry and costs for grid connection (e.g., Swider et al., 2008).

The Baltic Sea is in many ways ideal for establishing new offshore wind power farms and with nine countries surrounding the semi-enclosed sea there are many stakeholders. Today, mostly the southwestern parts of the basin have been utilized for

wind energy purposes (mainly by Denmark and Germany) but also other countries have well advanced plans for wind power expansion in the Baltic Sea in the coming decades (e.g., SWEA, 2019).

In general, forecast performance is better offshore than onshore as the sea surface is rather homogeneous and since diurnal cycles are less pronounced (Fennell, 2018). Still, forecasts struggle with accurate timing and magnitude of wind ramps (in connection to convective cells or passing front zones), extreme wind speeds, extreme wind shear and low-level jets (LLJs), a

phenomenon that is very common over the Baltic Sea during spring and summer (Kalverla et al., 2017; Tuononen et al., 2017; Hallgren et al., 2020).

With a close proximity to a coastline almost everywhere in the Baltic Sea there are many frequently occurring mesoscale meteorological events affecting the wind conditions to a varying extent, such as the sea breeze/land breeze circulation, LLJs and pronounced internal boundary layers (Svensson, 2018). Also, upwelling and wave-air interaction affects the lower part of

the wind profiles (Sproson and Sahlée, 2014; Wu et al., 2020).

In order to create an accurate forecast, usually numerical weather prediction (NWP) models are applied using clusters of supercomputers to calculate the evolution of the weather, although other methods also exist (see e.g., Hanifi et al., 2020). As a starting point for the calculations in an NWP model, a background field of assimilated initial conditions with observations from multiple sources is used. The quality of this field is crucial for the quality of the forecast, together with the model setup

regarding horizontal and vertical resolution as well as the parameterizations used to describe different physical processes. To improve the forecasts further, different post-processing techniques can be applied to the output from the NWP model, see e.g. Vannitsem et al. (2020) for a recent review.

This study presents a comparison of six commonly used post-processing methods to improve short-term deterministic wind power production forecasts for a site in the Baltic Sea, using data from an operational high-resolution NWP model and compar-

ing with observations from a LiDAR (Light Detection And Ranging). To provide a deeper understanding of the characteristics





that distinguish the methods, they were all implemented in their basic form. The performance of the methods under different conditions were evaluated together with an overview of their general performance.

As of today, one of the most popular post-processing methods is to apply machine learning (ML) algorithms to the NWP data. The number of studies on different ways to implement ML methods and their relative strengths and weaknesses have grown rapidly during the last decade (see e.g., Foley et al., 2012; Treiber et al., 2016). Common for all ML methods is that, using supervised or unsupervised learning, the algorithm finds patterns in the data and uses these patterns to make predictions based on data that was not in the training set. Among the most popular ML algorithms applied to post-processing of NWP data are neural networks, support vector regression algorithms and random forests (RF). In this study, we limited ourselves to test only the RF to highlight the benefits and limitations of using ML as a post-processing method. This particular method has been successfully applied to improve wind power forecasts by e.g. Lahouar and Slama (2017); Vassallo et al. (2020).

The paper is structured as follows. After the introduction in Sect. 1, a description of the site location, the LiDAR observations and the NWP model is presented in Sect. 2. In Sect. 3 the six post-processing methods are described together with a brief overview of the metrics applied to evaluate forecast performance. The results are presented in Sect. 4 and followed by a discussion in Sect. 5. A summary and concluding remarks can be found in Sect. 6. To simplify for the reader, a list of all abbreviations used throughout the text is presented at the end of the paper.

## 2 Materials

### 2.1 LiDAR measurements at Utö

Utö is a small island (approximately 1 km$^2$ in area) in the Baltic Sea, 60 km southwest off the coast of mainland Finland, see Fig. 1. The island is located at the southern edge of the archipelago and the nearest islands of similar size are approximately 12 km to the east and west of Utö, respectively, while to the south the sea is open (Tuononen et al., 2017). As part of the Finnish ground-based remote-sensing network, Utö hosts a scanning Doppler LiDAR and a number of other instruments allowing measurements of e.g. concentration and fluxes of greenhouse gases (Hirsikko et al., 2014).

The Doppler LiDAR at Utö is a fully scanning Halo Photonics Stream Line pulsed Doppler LiDAR, which was upgraded with an XR series amplifier and data acquisition in 2017. Halo Stream Line is a 1.5 $\mu$m pulsed LiDAR with coherent detector (Pearson et al., 2009) configured with 30 m range resolution and 7 s integration time per beam. Here, similar to Hallgren et al. (2020), we utilized only horizontal winds retrieved from a 15° elevation angle velocity azimuth display scan, which was configured with 24 azimuthal directions and operated every 15 min. The instrument was located 8 m above sea level and horizontal winds were retrieved from 35 m above sea level up at 7.8 m vertical resolution. In this study, only measurements on the 36 lowest height levels (35 to 307 m above sea level) were used. Before wind retrieval, measurements were post-processed according to Vakkari et al. (2019) and the radial data were filtered with a signal-to-noise ratio threshold of -23 dB.

Based on the LiDAR wind profile, the wind speed at hub height for a wind turbine (see Sect. 2.3) was calculated using a piece-wise cubic Hermite interpolating polynomial (PCHIP) fitted to the profile using logarithmic height coordinates (Fritsch and Carlson, 1980; Brodlie and Butt, 1991). Only 10 min averages with time stamps at even hours (hh:00) were used in the



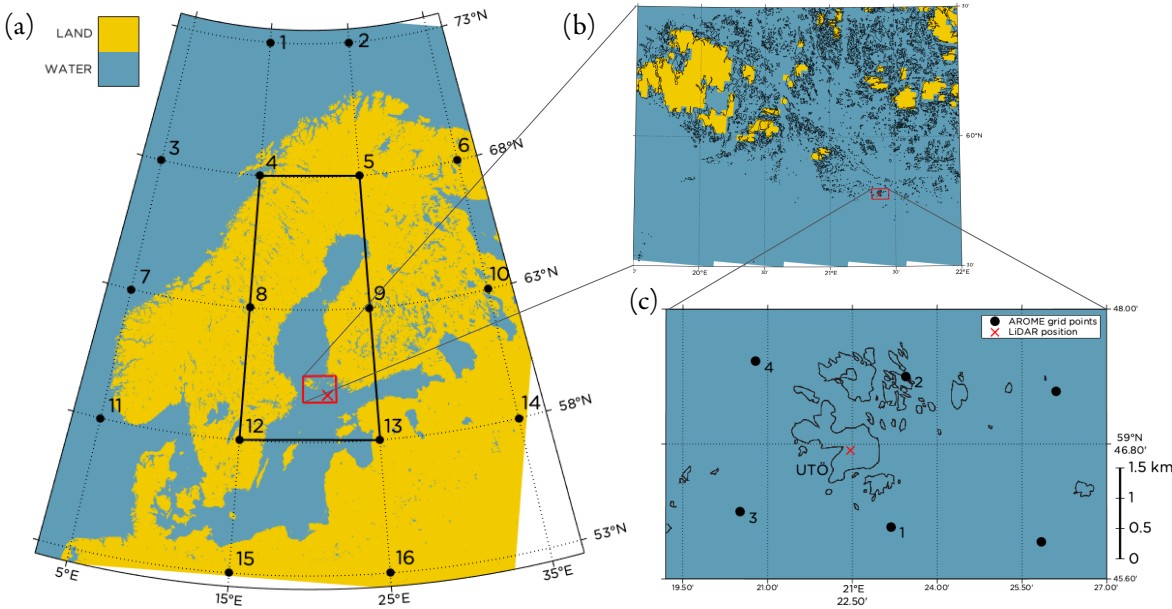

**Figure 1.** Land/sea-mask from AROME (Sect. 2.2) over (a) the Baltic Sea region, (b) the Finnish archipelago and (c) the surroundings of Utö. The position of the LiDAR is marked with an ×. In (a) the 16 grid points used for the JC method (see Appendix A) to classify the LWTs are marked, as well as the focus area for the classification. In (c) the four grid points closest to the LiDAR are marked and labeled according to their distance to the LiDAR, with 1 being the closest grid point.

analysis to get as instantaneous values as possible and to allow for a fair comparison with the NWP data (Sect. 2.2). The data

set covered a time period of two years, from February 1st 2018 to January 31st 2020. In this period, sea ice coverage extended to Utö only during parts of February, March and early April 2018.

Main criteria for data removal were positive and negative spikes in the profile. Also, if more than 75% of the data in a profile was missing, the profile was discarded. Furthermore, the LiDAR data was compared with measurements from a nearby meteorological tower at Utö and some observations were removed on manual inspection. In total 1.3% of the data was removed

in the quality control. Data availability is illustrated in Fig. 2. Throughout the study the LiDAR observations were used as the true values, not taking uncertainties of the measurements into account.

## 2.2 AROME forecasts

Deterministic forecasts from the HARMONIE-AROME (hereafter only called AROME) model system was used in this study. AROME is a high-resolution (2.5 km × 2.5 km and 65 vertical levels) convection-permitting atmospheric NWP model opera-

tionally used for short range weather forecasting by a number of countries around the Baltic Sea, including Sweden, Denmark, Finland, Estonia and Lithuania (Bengtsson et al., 2017). The domain covers Scandinavia and the Nordic Seas and most of the domain is shown in Fig. 1a. The AROME model used in this study was based on HARMONIE version cy40h1.1, see Bengts-



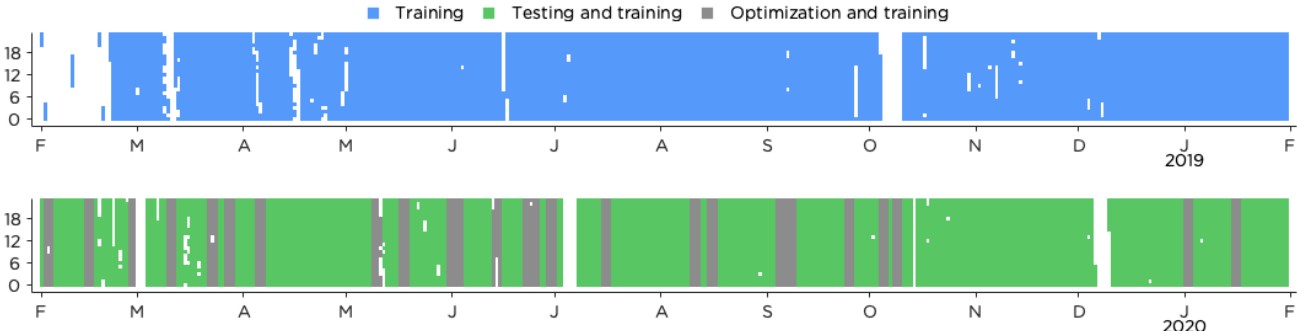

**Figure 2.** Availability of LiDAR observations (after quality control) and forecasts from AROME for the period February 1st 2018 to January 31st 2020. The time period is split into one year that was only used for training (February 1st 2018 to January 31st 2019) and one year that was used for training, optimization and testing (February 1st 2019 to January 31st 2020).

son et al. (2017) for more details on the model setup. Boundary conditions were from the European Centre for Medium-Range Weather Forecasts (ECMWF) using the Integrated Forecasting System (IFS) and 4D-var data assimilation. The AROME model
was run at 00, 06, 12 and 18 UTC with a data assimilation cycle of three hours. As only the effect and performance of different post-processing methods were of interest in this study, AROME was used only as a reference. We did not consider the performance of the NWP model itself nor compared it with other NWP models that potentially could have performed better (such as e.g. the high-resolution forecast from ECMWF). For a comparison of AROME and the ECMWF forecasts we refer to Müller et al. (2017) and Kalverla et al. (2019).

In this study only forecasts initialized at 00 UTC with forecasts lengths of 0–23 hours (D1) in hourly time steps were used. The forecasts were assumed to be available at 00 UTC without delay. For some post-processing methods also the 24–47 (D2) hour forecasts were used (see Sect. 3.2). Data was retrieved for the four grid points closest to the LiDAR, as marked in Fig. 1c. The distances between the LiDAR and the grid points were all in the range 1.4 to 2.2 km.

Horizontal wind components and temperature at the 11 lowest model levels (reaching up to approximately 320 m above sea
level) were retrieved. By fitting a PCHIP to the profile, the wind speed at hub height was calculated in the same way as for the LiDAR observations (Sect. 2.1). The wind direction at hub height was calculated using linear interpolation between the wind components at the two model levels closest to the hub height, and similarly the temperature at hub height was calculated. Specific humidity and air pressure at the two lowest model levels (approximately 12 and 37 m above sea level) was retrieved and the bulk Richardson number in this layer was calculated based on information on temperature, pressure, humidity and
wind speed (Stull, 2012). Additionally, sea level pressure for the 16 grid points marked in Fig. 1a was retrieved to allow for calculation of the Lamb Weather Types (LWTs) (Lamb, 1972; Jenkinson and Collison, 1977).

Only time steps when forecasts and LiDAR observations were available simultaneously were used, see illustration of data availability in Fig. 2.





## 2.3 Power curve

Forecasting the actual power production from a wind farm faces many challenges such as accurately predicting the wind speed and the density of the air as well as production losses related to e.g. icing on the wind turbine blades (Lamraoui et al., 2014; Molinder et al., 2021) and wake effects (Shaw et al., 2009; Butler, 2014). To only focus on the contribution from the wind, a theoretical maximum production from a wind turbine at the location of the LiDAR was calculated using the power curve for a Siemens SWT-3.6-120 wind turbine (Siemens, 2011). This particular type of turbine is commonly used in the Baltic Sea today,

with 111 turbines in the Anholt farm (commissioned 2013) and with 80 turbines in the EnBW Baltic 2 farm (commissioned 2015). The turbine has a hub height of 90 m and the blades sweep heights from 30 to 150 m. The cut-in wind speed is 3.5 m s$^{-1}$ and the turbine reaches its rated production of 3.6 MW at 14 m s$^{-1}$. The cut-out wind speed is 25 m s$^{-1}$.

## 3 Evaluation metrics and methods for post-processing

In this section the different metrics used to evaluate the performance of the post-processed forecasts are presented, together

with a detailed description of the different methods applied.

### 3.1 Evaluation metrics

The deterministic 0–23 h AROME forecast was used as a baseline for the statistics. As the key metric we selected the mean absolute error (MAE) skill score defined as

$$\text{MAE}_{\text{skill score}} = 1 - \frac{\text{MAE}_{\text{new}}}{\text{MAE}_{\text{det}}} \tag{1}$$

where $\text{MAE}_{\text{new}}$ and $\text{MAE}_{\text{det}}$ are the mean absolute errors from the new (post-processed) and original deterministic forest respectively. All forecast lengths (0–23 h) were treated equally as no substantial decrease in forecast performance over D1 was expected. A MAE skill score of 1 indicates a perfect forecast ($\text{MAE}_{\text{new}} = 0$) while a skill score of 0 means that the new forecast had the same skill as the original forecast. A negative skill score implies that the post-processing had deteriorated the original forecast quality and that the new forecast had worse performance.

As a complement to the MAE skill score also the forecast bias is presented and the correlation coefficient, standard deviation and centered root mean square error (CRMSE) are visualized in a Taylor diagram (Taylor, 2001). To compare frequency distributions for forecast improvements the Earth mover's distance (EMD) was used to objectively asses how similar the distributions were to an optimal distribution (Rubner et al., 2000). The EMD is equal to the area between the cumulative distribution functions and can be conceptualized as the minimum of work needed to transform one distribution into the other.

Accompanying the frequency distribution for forecast improvements is also the forecast superiority score which reveals how often it would have been better to use the new forecast rather than the original forecast.





## 3.2 Methods for post-processing

Six different post-processing methods were applied to the AROME forecasts, most of them using a measure-correlate-predict approach. To facilitate evaluation of different aspects of the results of post-processing, the methods were kept as simple as possible. All methods tested are easy to implement and are commonly used in operational power production forecasting. The methods were also selected on the basis of cost-efficiency (in terms of computational time) and all can run on a laptop. As ML is a rapidly advancing field of post-processing with many different approaches possible, we selected only one of the most common methods (the random forest) to demonstrate the benefits of ML.

All methods were applied and tested in the same manner as they would have been if used operationally, except that the forecasts from AROME were assumed to be available immediately at 00 UTC, which would not be the case in operational use where the delay is approximately 3 h. Also, at 00 UTC only observations up until 23 UTC from the day before were assumed to be available. Thus, the most recent observations were always at least one hour old.

The two years of data were divided into one year (February 1st 2018 to January 31st 2019) that was used for training only and one year (February 1st 2019 to January 31st 2020) that was used for training, optimization of the RF algorithm and for testing the performance of the methods. The optimization period consisted of 20% of the second year, randomly selected in blocks of three days (see Fig. 2) to minimize problems with auto-correlation in the data. For all methods, only historical data up until the first time step of the current forecast was used as training data, simulating a realistic power production forecasting methodology. The first day in the training data was always February 1st 2018 unless otherwise stated, and hence the amount of training data grew by one day for each new forecast.

In the following subsections, the six different post-processing methods are described in detail.

### 3.2.1 Forecast from yesterday (D1/D2 MIX)

The AROME forecast issued at 00 UTC from the day before with a lead time of 24–47 hours (D2) was compared to the current 00 UTC AROME forecast valid for the coming 0–23 hours (D1). The wind speeds from the two forecasts were combined into a common forecast with weights based on their performance in terms of absolute error in power production during the test period.

### 3.2.2 Persistence forecast (Pers)

A persistence forecast is typically used as a reference forecast to study forecast improvements over a short time span (see e.g. Nielsen et al. (1998); Soman et al. (2010)). In our case, a persistence forecast was generated assuming that the most recent (from 23 UTC) wind speed measured by the LiDAR would remain constant throughout the following day. Thus, the persistent power production would also remain constant. If observations from 23 UTC were missing, no forecast was created. This was the case for four days during the test period.

### 3.2.3 Neighbourhood method (NBH)

For the NBH method, the D1 forecasts from the four grid points closest to the position of the LiDAR (see Fig. 1) were combined into a common wind speed forecast. The forecasts were weighted depending on their performance in terms of the absolute error

in power production during the training period for eight wind directions (45° per sector). The wind direction was calculated as the average wind direction at hub height from the four grid points.

### 3.2.4 Temporal smoothing (Smooth)

The D1 forecast was smoothed in time by applying a low pass filter (moving average) calculating the average wind speed at every forecast lead time (0–23 h) using forecast data within a window of ±1 hour. For the first time step in the forecast,

00 UTC, the average was based on the LiDAR observations at 23 UTC together with the forecast data for 00 UTC and 01 UTC.

### 3.2.5 Linear regression (LR) methods

A set of four different methods applying linear regression (minimizing the error in wind speed between the D1 forecast and the LiDAR observations in terms of least squares) were tested using different criteria to split the training data into non-overlapping subsets.

**LR all**

In the most basic LR method we fitted a first order polynomial to all data in the training set and corrected the D1 forecast accordingly. As the training data set grew by one day for each new forecast, the polynomial was refitted every day to include the new data.

**LR stability**

Using data from the two lowest model levels in AROME the bulk Richardson number, $Ri_{bulk}$, was calculated. Based on the classification by Lee et al. (2017), only changing the threshold for strongly unstable stratification, the following five stability classes were used:

  – strongly stable if $Ri_{bulk} \geq 0.25$

  – stable if $0.05 \leq Ri_{bulk} < 0.25$

– neutral if $-0.05 \leq Ri_{bulk} < 0.05$

  – unstable if $-1 \leq Ri_{bulk} < -0.05$

  – strongly unstable if $Ri_{bulk} \leq -1$



For every new forecast all time steps in the training data were split into five groups depending on their stability. Note that also for the training data, stability from AROME was used. A first order polynomial was fitted to the wind speed data for each
stability class and the different calibrations were applied to the D1 forecast depending on the forecasted hourly stability.

**LR synoptic**

To examine the synoptic conditions, the JC method (Jenkinson and Collison, 1977) was used to calculate a reduced set of 11 LWTs, see Appendix A for details. The LWTs describe the synoptic weather pattern over the Baltic Sea and can be either cyclonic (C), anticyclonic (A), flow from any of the eight wind directions (N/NE/E/SE/S/SW/W/NW) or unclassified (weak)
flow (U) (Lamb, 1972). Similar to the LR stability method, the LR synoptic method classified every time step in both the training data and in the D1 forecasts and applied linear regression to adjust the forecasted wind speed based on the forecasted LWT.

**LR seasonal**

To address seasonality in forecast performance the training data was split into three seasons: late spring/early summer (April –
July), late summer/fall (August – November), winter/early spring (December – March). Following the same procedure as for the other LR methods, the D1 forecast was adjusted using a first order polynomial fitted to the wind speed training data for the current season.

### 3.2.6 Random forest (RF)

The RF method is a commonly used strategy to increase forecast performance through ML post-processing of NWP data. The
technique has been successfully applied to different aspects of wind power production forecasting before, both onshore and offshore, see e.g. Lahouar and Slama (2017); Vassallo et al. (2020).

A random forest (Breiman, 2001) creates a group (a forest) of individual decision trees. Based on a random selection of the training data, each tree outputs a prediction of the wind speed or wind power production. The average of the predictions from all trees then holds as the final prediction for the RF.

As individual decision trees are prone to overfitting, the random selection of training data given to the forest of decision trees minimizes this problem. Here we used the bootstrap aggregated random forest TreeBagger as implemented in MATLAB 2018a (MathWorks, 2021). To allow the RF to find patterns in the data and correct the forecast simultaneously for all aspects that were possible for the less complex methods, the full training data set contained the same information[1] that was available to the other methods together with some additional information such as temperature at hub height. The complete list of the 19
training features that were available for the RF are presented in Table 1.

Feature selection can be performed in many different ways and there are several studies discussing the best approach, see e.g. Kursa and Rudnicki (2010) and Cai et al. (2018). In order to fully understand the relative importance of the different training

---

[1]Instead of using the sea level pressure from the 16 grid points used in the JC method the derived synoptic vorticity $Z$ (see Appendix A) was used. Together with information about wind direction it provides the same information as was used for LWT classification.



**Table 1.** Training features available for the RF algorithm. All forecasts for wind speed, wind direction and temperature are valid for hub height (90 m). The indices denote the grid points as marked in Fig. 1c. The persistence forecast for wind speed was generated using the LiDAR observations at 23 UTC as described in Sect. 3.2.2.

| | |
|---|---|
| wind speed 1 D1 | wind speed 1 D2 |
| wind speed 2 D1 | wind speed 2 D2 |
| wind speed 3 D1 | wind speed 3 D2 |
| wind speed 4 D1 | wind speed 4 D2 |
| wind direction 1 D1 | wind direction 1 D2 |
| temperature 1 D1 | temperature 1 D2 |
| $Ri_{bulk}$ 1 D1 | $Ri_{bulk}$ 1 D2 |
| $Z$ D1 | $Z$ D2 |
| persistence wind speed | |
| hour of day | |
| day of year | |

features, we performed feature selection using a step-by-step optimizing chain. First, the random forest was trained using all the 19 training features individually and using standard settings for the random forest (see below). The training feature that

gave the best performance in terms of MAE skill score for power production for the optimization period (Fig. 2) was kept. For the second round in the chain, the remaining 18 training features were tested in combination with this feature. The process was repeated, adding the feature that gave the maximum increase in MAE skill score for each round in the chain, until there was no further improvement. This optimal set of training features was then kept constant as the number of trees in the forest were changed. Forest sizes of 10, 50, 100, 150, 200 and 250 trees were tested with 50 trees being the standard selection.

Most ML algorithms have a set of hyperparameters that are used to control the learning process and adjust the algorithm to the problem. In the case of TreeBagger, the hyperparameter Minimum Leaf Size (MLS) refers to the minimum number of observations per leaf in a decision tree, and thus is inversely related to the number of branch splits in a tree. The default setting is 5 for regression problems, but MLS of 1 and MLS of 10 to 50 in steps of 5 were aslo tested. The response of changing the MLS was tested for the optimal set of features and the optimal number of trees and was evaluated in terms of MAE skill score

for power production for the optimization period.

The RF was implemented in three different ways:

- – wind speed to wind speed (ws to ws): using the training features described in Table 1, the RF was optimized using the wind speed from the LiDAR as response data

- – wind speed to power production (ws to pwr): using the training features in Table 1, the RF was optimized using the

calculated power production from the LiDAR measurements as response data



– power production to power production (pwr to pwr): converting the wind speed features in Table 1 to power production, the RF was optimized using the power production calculated from the LiDAR measurements as response data.

Out of these three setups, the one with the highest MAE skill score for the optimization period was then selected for comparison with the other post-processing methods.

# 4   Results

In this section, the general meteorological conditions and theoretical power production at Utö during the two years of measurements and forecasts are presented, followed by an overview of the performance of the original and the post-processed forecasts. Also, the forecast performance in different weather situations is presented. For the RF, details about the optimization of the features and settings in the algorithm are given.

## 4.1   Meteorological conditions

An overview of the LiDAR wind speed at 90 m hub height is presented in Fig. 3a together with the theoretical power production if a SWT-3.6-120 would have been placed at the site and assuming that the power production was following the power curve perfectly (Fig. 3b). The monthly average wind speed was in the range 6.7–12.6 m s$^{-1}$ with the lower values during the summer months and the higher wind speeds during winter. The deterministic forecast error in terms of CRMSE was somewhat higher
during spring and summer (April–July) for both years. Similarly, the average power production reached its peaks during the winter months, with the largest forecast errors during spring and summer (Fig. 3b). Also, January 2019 displayed errors of similar size. Note that the high values of CRMSE (both for wind speed and power production) for February 2018 might be due to the small number of data points in this month as there were a lot of missing observations (see Fig. 2).

The monthly distribution of atmospheric stratification classes, based on the bulk Richardson number from the AROME
forecast and the classification presented in Sect. 3.2.5, is shown in Fig. 4a. Both years were similar with stable stratification being the dominant class during spring and summer (April–July). February, March and August were transition months while the rest of the months were mostly unstable or strongly unstable.

Using the JC method to calculate a reduced set of 11 LWTs (see Appendix A), the monthly distribution of the synoptic weather patterns is presented in Fig. 4b. It is clear from the graphics that anticyclonic (A) and cyclonic (C) weather types were
dominating together with winds from south to west, but variations between the months are noticeable. As mentioned earlier, note that the result for February 2018 is based on a much smaller number of data points.

The scatter plots for wind speed and power production in Fig. 5 show that both years were similar. Thus, the first year was a representative training set that could be used to improve the forecasts for the second year. Although most of the data points were located close to the 1:1 line it was not uncommon that forecast errors reached 5 m s$^{-1}$ or more (1.2% of the time). For
power production, the spread was large for intermediate wind speeds where the power curve is steepest as small errors in forecasted wind speed here get amplified in forecasted power production. For example, an error of 0.1 m s$^{-1}$ at 9 m s$^{-1}$ would



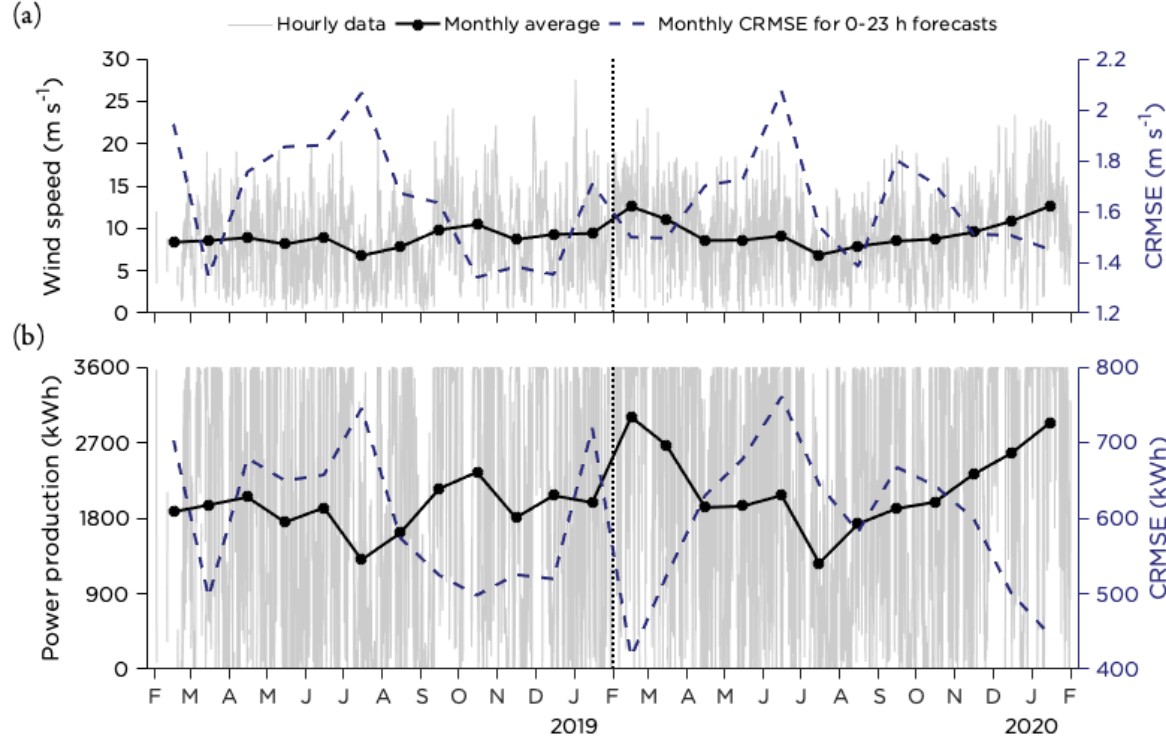

**Figure 3.** Overview of hourly data and monthly averages of (a) wind speed and (b) theoretical power production at Utö during the two years of measurements. Also CRMSE for wind speed and power production forecasts from AROME are presented.

result in a power production error of 70 kWh. In contrast to this, the forecasts performed better in terms of power production for both high and low wind speeds as expected, since sensitivity to forecast errors is lower in those ranges.

## 4.2 Optimizing the RF

The optimization period, see Sect. 3.2, was used to find the optimal set of training features and settings for the RF method. Using standard settings for the RF, the training features were added one by one in a process described in detail in Sect. 3.2.6. The features ordered according to their importance for the MAE skill score for the optimization period are presented in Table 2. The full set of training features tested are given in Table 1 and we note that the temperature and synoptic vorticity $Z$ were not used by any of the three setups. Also, Table 2 indicates that data from grid point 4 seems to be the most relevant. The increase
of MAE skill score for the optimization period when the features were added as listed in Table 2, is presented in Fig. 6a.

Figure 6b shows how the MAE skill score changed with the number of trees (keeping MLS constant at 5) and Fig. 6c the response when changing MLS (keeping the number of trees constant at the value found in Fig. 6b). For the RF setup "wind speed to wind speed" the optimal settings were 150 trees and an MLS of 5. RF "wind speed to power" (RF "power to power") was optimized using 200 (250) trees and an MLS of 5 (25). The effect of changing the training length is shown in Fig. 6d,

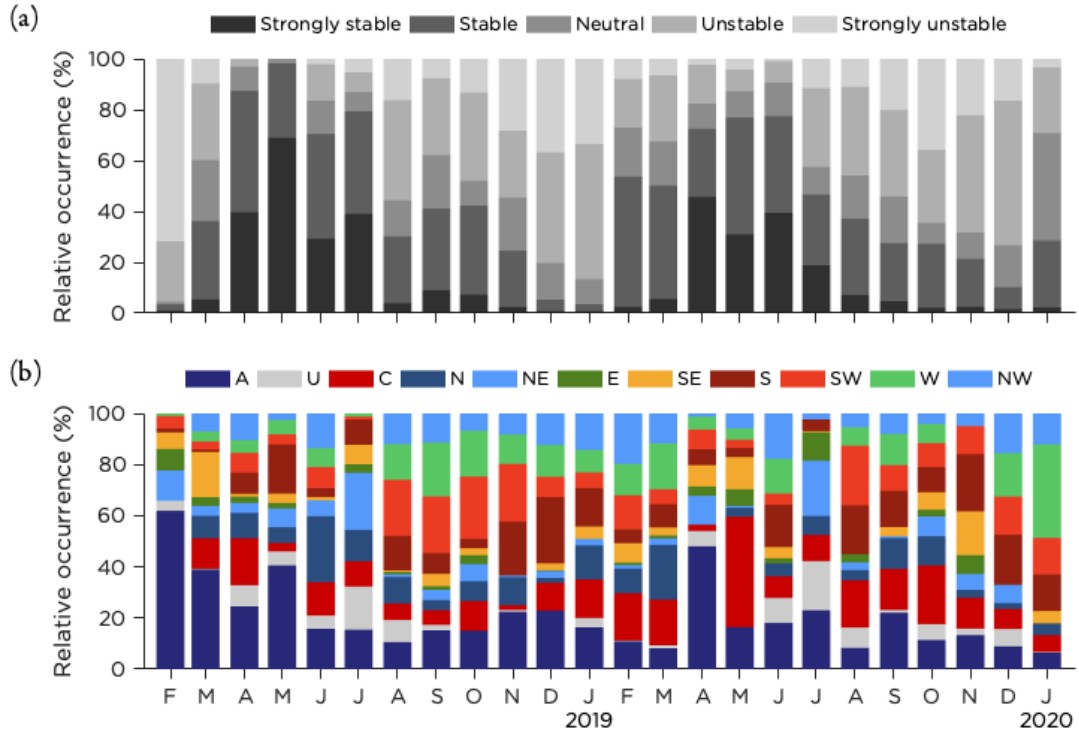

**Figure 4.** Monthly occurrence rate of (a) the boundary layer stratification based on data from the two lowest model levels in AROME forecasts and (b) the synoptic situation using a reduced set of 11 LWTs during the time period studied.

using the 10 most recent days up to the full training period that was extended by one day for each new 0–23 h forecast, at least including 365 days (forecasting for February 1st 2019) and at the most 729 days (forecasting for January 31st 2020). As seen in the figure the performance changed drastically going from 10 to 50 trees but showed only minor variations for 50–250 trees. Regarding MLS, all setups indicate lower MAE skill score for an MLS of 1. Increasing the MLS from 5 to 50 decreased the performance for the setup RF "wind speed to power" but was more or less constant for the other two setups. When it comes to

training length, the more training data, the better the performance of the RF.

The best performance in terms of MAE skill score for the optimization period was achieved using RF "wind speed to wind speed" with the eight features given in Table 2, 150 trees, a MLS of 5 and the maximum training length. This setup is hereafter referred to only as RF.

### 4.3 Overall performance of the post-processing methods

The overall performance for all post-processing methods tested is shown in Fig. 7. Only the NBH method, smoothing and the RF manage to slightly increase the MAE skill score with smoothing giving the skill score of 0.045. The smoothing method was also the method with least bias. Most of the different post-processing methods are clustered in the Taylor diagram (Fig.



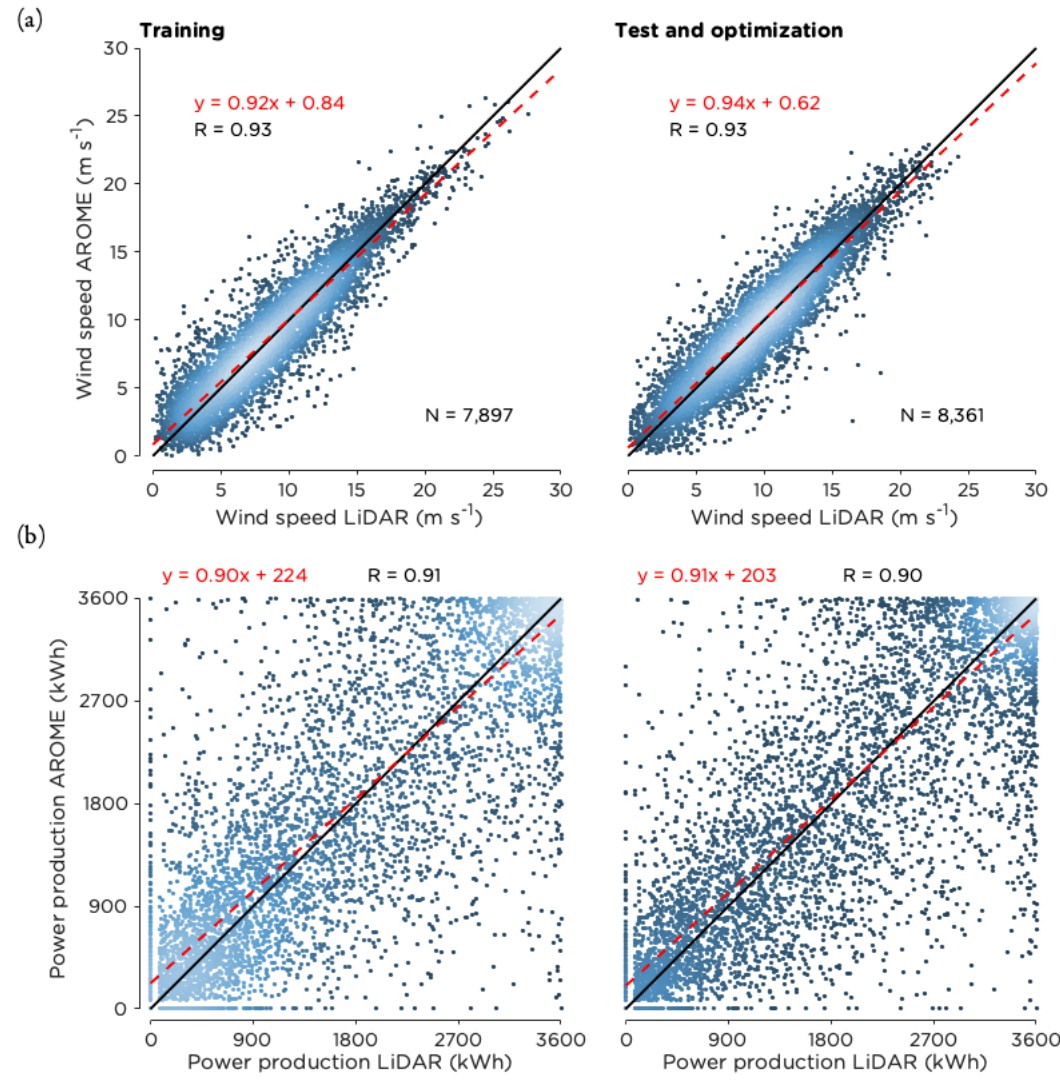

**Figure 5.** Scatter plots of (a) wind speed at hub height and (b) power production comparing the LiDAR data with the deterministic D1 forecasts from AROME during the two years analyzed. The black line is the 1:1 ratio and the dashed line in red is the best linear fit to the data. The equation for the best fit is given together with the correlation coefficient (R) and the number of data points (N). The coloring of the data points indicate the density of the data with brighter colors representing higher density.



**Table 2.** Feature appearance in order of selection based on improvement of MAE skill score for the optimization period for the three tested RF setups.

|   | ws to ws | ws to pwr | pwr to pwr |
|---|---|---|---|
| 1 | wind speed 4 D1 | wind speed 4 D1 | power 4 D1 |
| 2 | wind speed 1 D2 | wind speed 3 D2 | power 1 D2 |
| 3 | wind direction 1 D2 | wind direction 1 D2 | wind direction 1 D2 |
| 4 | $Ri_{bulk}$ 1 D2 | wind speed 3 D1 | hour of day |
| 5 | hour of day | $Ri_{bulk}$ 1 D1 | $Ri_{bulk}$ 1 D1 |
| 6 | persistence wind speed | hour of day | power 2 D1 |
| 7 | wind speed 2 D1 | – | persistence power |
| 8 | day of year | – | – |

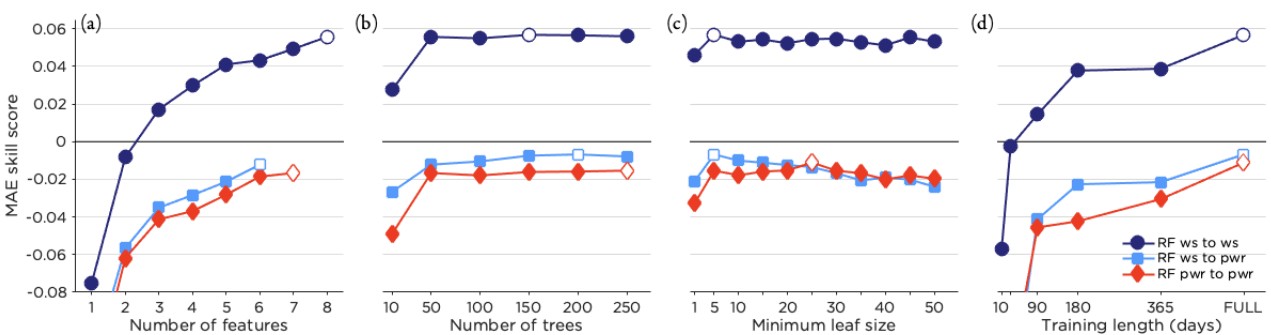

**Figure 6.** MAE skill score for the optimization period for the three RF setups tested. In (a) the improvement of MAE skill score when training features were added according to the order in Table 2 is shown. Panels (b), (c) and (d) show how the MAE skill score was affected by changing the number of trees (using the best set of features), the MLS (for the best set of features and optimal number of trees) and the training length (for optimal selection of features, number of trees and MLS). The optimal setting for each setup is marked in white. Note that the values for MAE skill score presented here are not directly comparable to the MAE skill scores presented in other figures in this paper as these are for the optimization and not the test period.





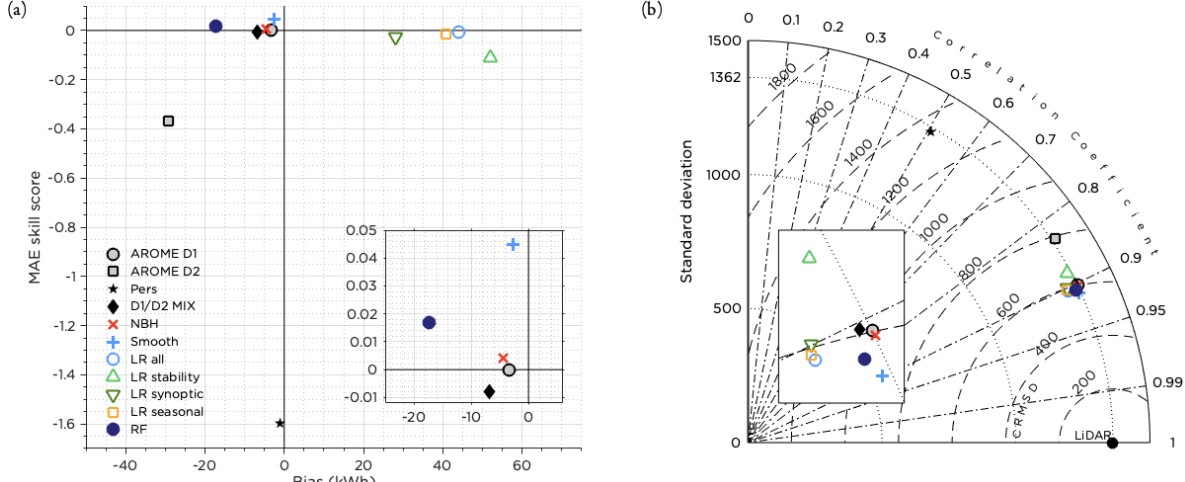

**Figure 7.** Performance of all post-processing methods during the test period. Panel (a) shows the MAE skill score and bias and (b) the Taylor diagram. The insets in (a) and (b) show enlarged portions of the figures to more clearly show the differences between the methods.

7b), but also here it seems that the smoothing method and the RF show the best performance, increasing the correlation and decreasing the CRMSE. While most methods decreased the variability of the forecasts, the NBH and smoothing methods

had the best conformity. It should also be mentioned that the original forecast was almost perfect with regard to variability. Using the D2 deterministic forecast (24–47 h lead time) or the persistence forecast gave negative skill scores and lowered the correlation coefficient drastically. However, using the D2 forecast in combination with the D1 forecast (D1/D2 MIX) might still be an option, even though the technique to combine the two forecasts tested here was not optimal. None of the LR methods succeeded in improving the forecast.

The frequency distributions in Fig. 8 show forecast improvement compared to the original forecast for the different post-processing methods, compared to the maximum possible improvement for a perfect forecast. All methods show a high peak for small changes compared to the original forecast, in most cases with more than 30% of the time steps within the ±25 kWh bin. The NBH method alters the original forecast the least, with 73% of the time steps in this bin.

The forecast superiority value answers the question how often it is better to use the post-processed forecast, counting only

hours with improvement greater than 25 kWh. For a perfect forecast these major improvements would occur 69% of the time. The RF had the highest forecast superiority score of 32% but failed to improve the forecast 28% of the time. The smoothing method, which was superior in 28% of the cases, was inferior to the original forecast 22% of the time. These two methods, together with the NBH method, are the only methods that managed to improve the forecast more often than they deteriorated it.

The EMD values presented in Fig. 8 provide information on the similarity of the distributions to that for the perfect forecast, the lower the value the more similar the distributions. As expected from earlier results, the smoothing method and the RF had the best EMD scores, while the persistence and the D2 forecasts were furthest away from the optimal distribution.





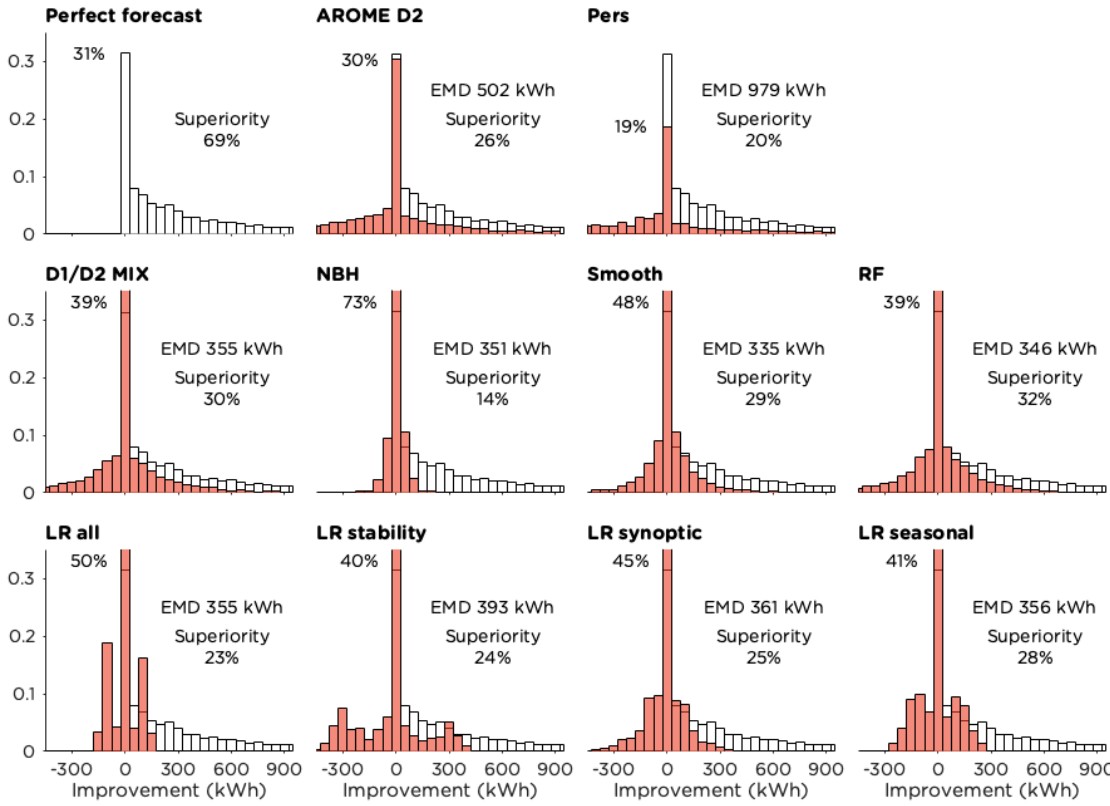

**Figure 8.** Frequency distributions showing the forecast improvement compared to AROME D1 in kWh for the different methods tested. The methods are compared to a perfect forecast (white bars) and the EMD values between the two distributions is given. The forecast superiority (improvements greater than 25 kWh) is presented and also the percentage of the time steps with minor alterations of the original forecast (changes less than ±25 kWh).

For comparison, adding Gaussian distributed noise with an average of 0 m s$^{-1}$ and a standard deviation of 0.5 m s$^{-1}$ to the original forecast resulted in an EMD value of 372 kWh and a forecast superiority of 26%. For this forecast, 42% of the data points were within ±25 kWh from the original forecast and the MAE skill score was -0.055.

### 4.4 Forecast performance in different meteorological conditions

Studying the forecast improvement for different wind speed bins based on the forecasted wind speed, Fig. 9, we see that smoothing improved the forecast for almost all wind speeds while the RF only resulted in improvement for intermediate wind speeds. This was also the case for the D1/D2 MIX. Most methods underestimated the power production for wind speeds up to 7.5 m s$^{-1}$ and overestimated the production for wind speeds above 10 m s$^{-1}$ (until the rated wind speed was reached). The errors were largest for intermediate wind speeds (Fig. 9b), related to the shape of the power curve.

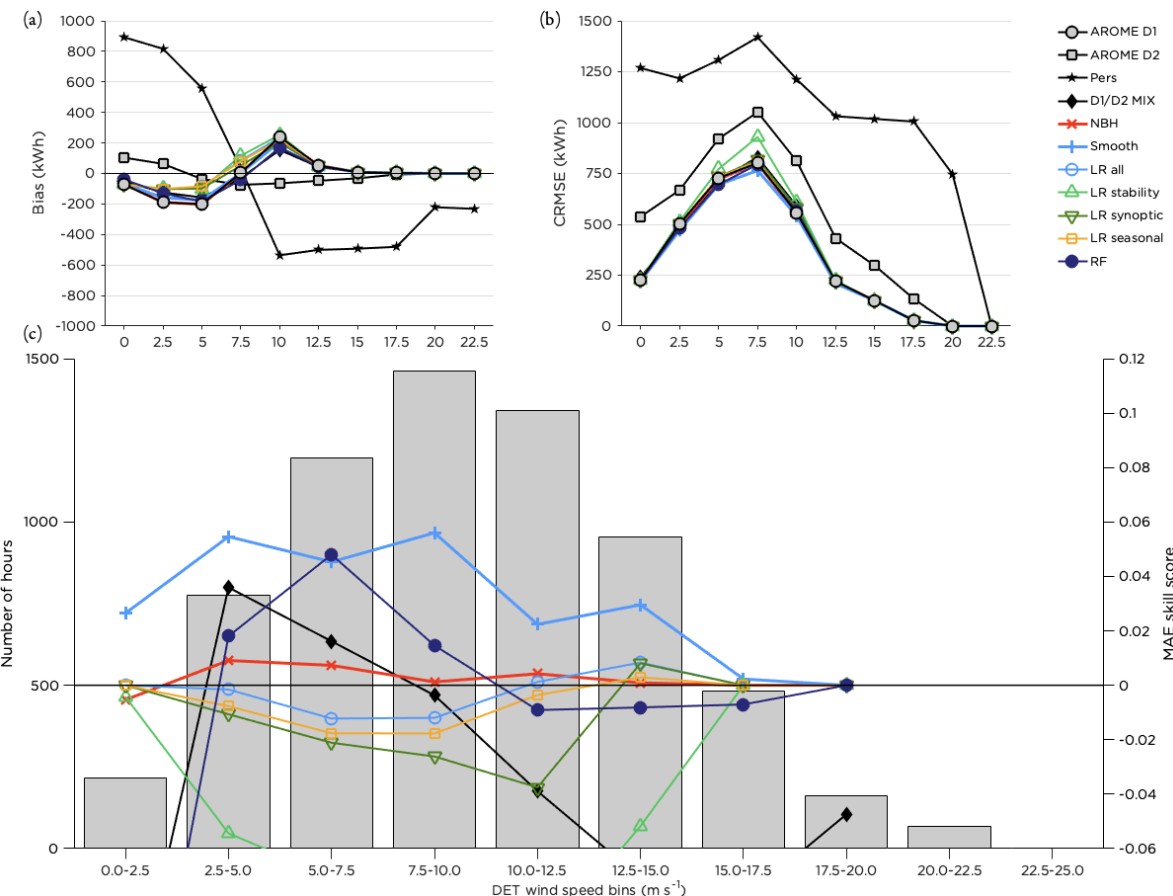

**Figure 9.** Performance of the post-processing methods for different wind speed bins. Panel (a) shows the bias, (b) the CRMSE and (c) the MAE skill score for the methods with the highest scores together with the distribution of the number of hours forecasted within each bin during the test period.

No clear pattern was visible when splitting the test data into hours of the day (not shown), except that the smoothing method gave a major improvement for the first time step (00 UTC) due to interpolation of the observations at 23 UTC. The use of observations was also the reason for the higher performance of the persistence forecast for this time step.

The smoothing method managed to improve the forecast in all stability classes and in all LWTs, see Fig. 10 and 11. Errors were generally larger for stable cases and lower for neutral stratification (Fig. 10b). Most methods, as well as the original forecast, underestimated the power production for strongly stable and strongly unstable conditions, while biases were smaller for less strong stratification (Fig. 10a). The RF performed well for stable and neutral cases but did not improve the forecast as much for time steps with unstable stratification (Fig. 10c). Interestingly, the D1/D2 MIX was the method that performed best

under neutral conditions. Relating to this, it can be noted in Fig. 11c that this method was superior in the unclassified (weak) flow. It also performed well in the 2.5–5.0 m s$^{-1}$ bin (Fig. 9c).

WIND
ENERGY
SCIENCE
DISCUSSIONS

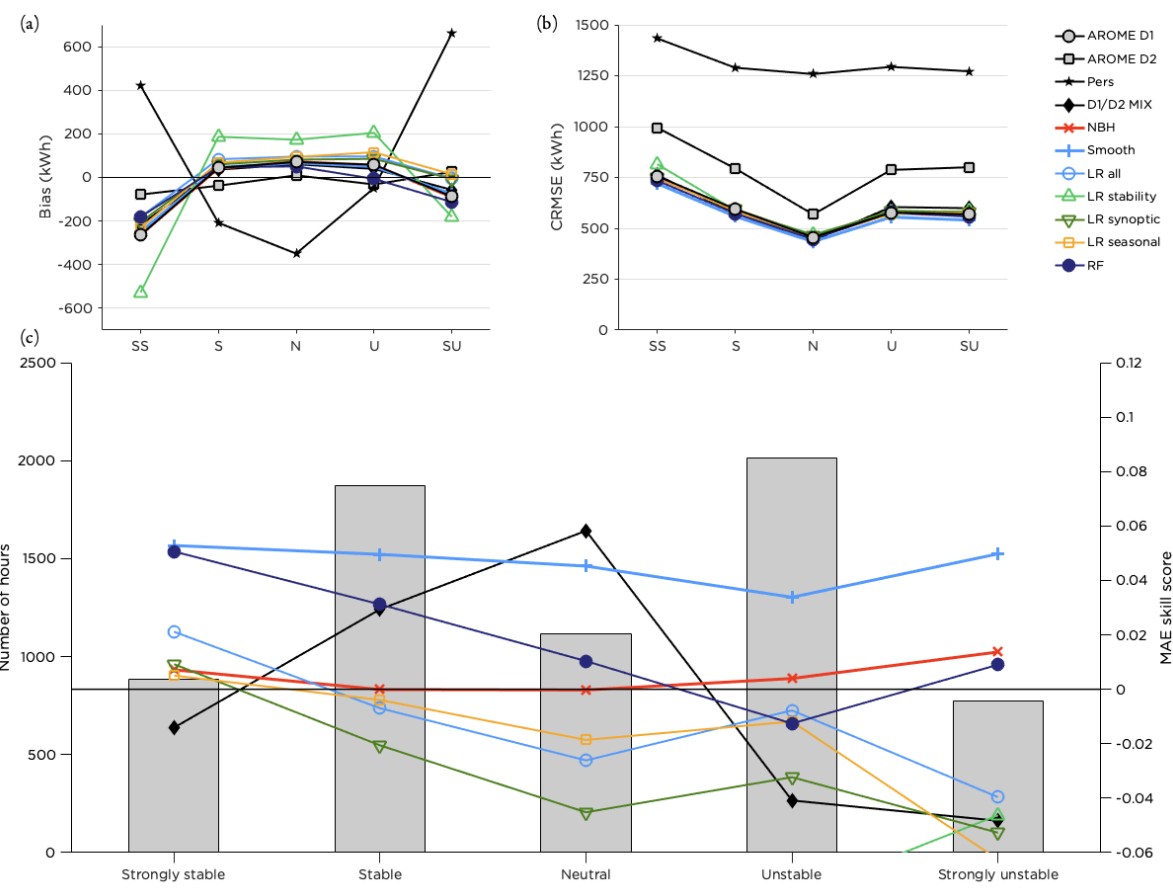

**Figure 10.** Performance of the post-processing methods for different stability classes. Panel (a) shows the bias, (b) the CRMSE and (c) the MAE skill score for the methods with the highest scores together with the distribution of the number of hours within each stability class during the test period.

The results for the performance in the different LWTs are similar to what could be seen in earlier figures with the smoothing method and RF on top for most synoptic situations. However, for easterly and northwesterly flow, the RF method did not improve the forecast. Figure 11c indicates that the performance of the RF was dependent on the amount of training data in 355 the class (note however that the bars represent the number of hours per class during the test period, not the training period). Forecast errors (Fig. 11b) are somewhat smaller for anticyclonic (A) and unclassified (U) flow and winds from the sector south to northwest (the most common wind directions) and slightly higher for pure cyclonic (C) flow and winds from north to southeast.

LLJs are frequently occurring over the Baltic Sea and with wind maxima on low levels they alter the ordinary logarithmic 360 or power-law wind profile. When a falloff criterion of $1 \text{ m s}^{-1}$, as defined in (Hallgren et al., 2020), was applied to the wind data on the 11 lowest model levels in AROME (spanning from approximately 12 to 320 m, see Sect. 2.2), LLJs were predicted

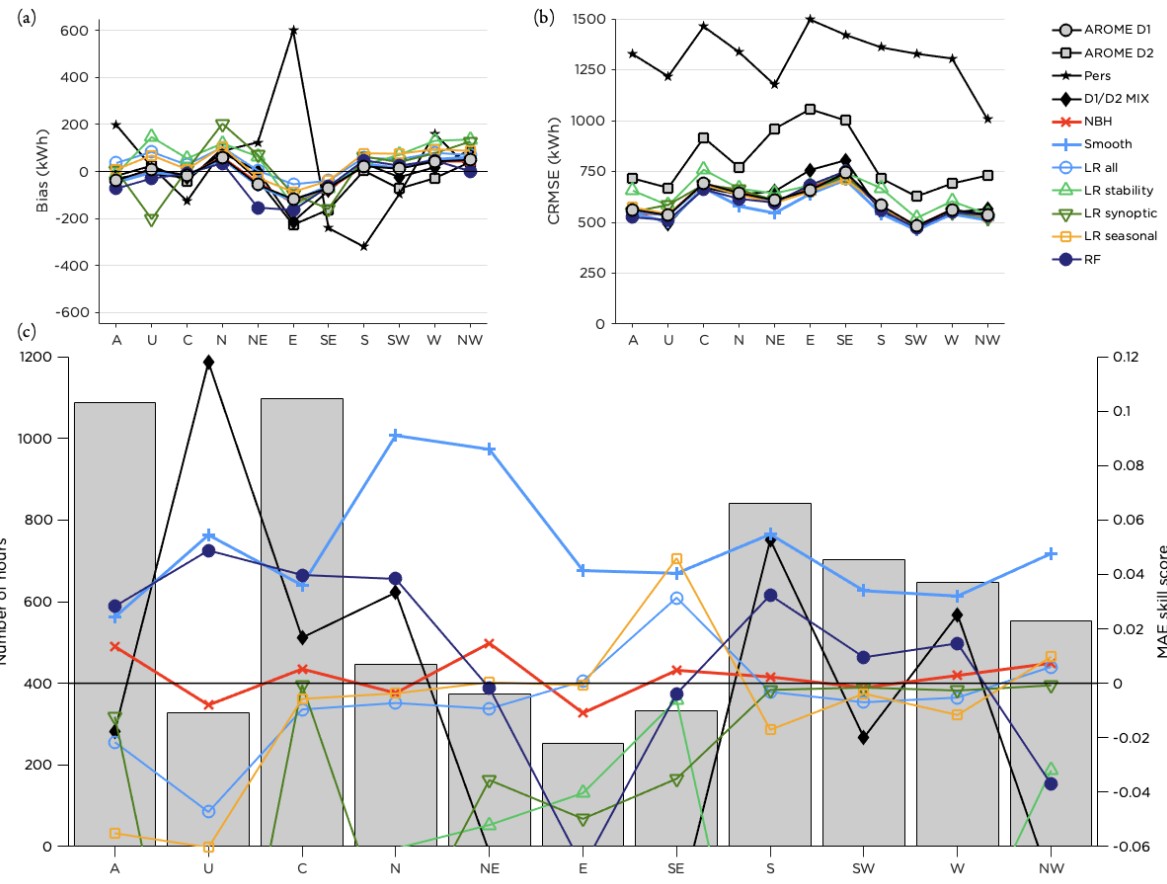

**Figure 11.** Performance of the post-processing methods for different LWTs. Panel (a) shows the bias, (b) the CRMSE and (c) the MAE skill score for the methods with the highest scores together with the distribution of the number of hours within each LWT during the test period.

during 13% of the time in the test period. Out of these events, 56% were correctly forecasted within the same time step as when an LLJ was identified in the LiDAR data using the same criterion. The total frequency of LLJs is somewhat underestimated by AROME, with the LiDAR observing LLJs 16% of the time in the test period (not shown).

The RF method performed better than the other post-processing methods when an LLJ was forecasted but produced no or negligible improvement when no LLJ, see Fig. 12. Also the D1/D2 MIX was successful during LLJs, but deteriorated the forecast quality otherwise. The performance of the temporal smoothing was less sensitive to whether a LLJ was present in the forecast or not, and improved the forecast in both cases.



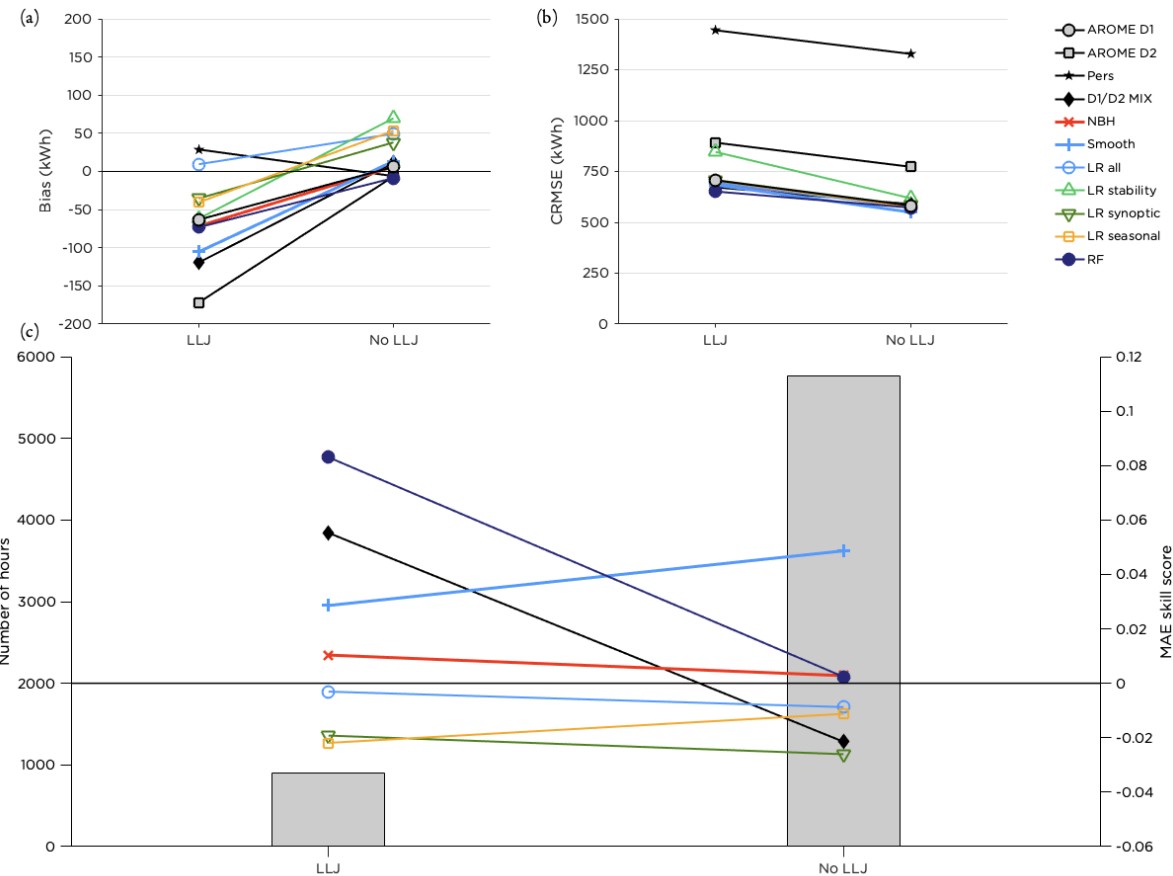

**Figure 12.** Performance of the post-processing methods for cases when an LLJ is forecasted compared to when there is no LLJ in the wind profile. Panel (a) shows the bias, (b) the CRMSE and (c) the MAE skill score for the methods with the highest scores together with the distribution of the number of hours with or without an LLJ in the forecast during the test period.

## 5 Discussion

Even though NWP forecasts for offshore wind power production are relatively good compared to forecasts for wind conditions in complex terrain, there is still room for improvement. The post-processing methods tested in this study are all commonly used and are implemented as they would be used operationally and in basic formulations, to allow a clean analysis of their respective advantages and disadvantages.

The methods tested are all general and could be applied to longer forecast lead times, to other variables than wind speed
and onshore as well as offshore. However, as most methods require a substantial amount of training they are sensitive to model updates and would need retraining from scratch in case of a major update that affects the NWP performance for the variable of interest. For AROME, there was a major update in February 2020, which is the reason that we did not include more recent forecasts. Also, if the observations are prone to systematic errors or large uncertainties that are not removed or minimized





through quality control, there is a substantial risk of making the forecast worse when applying the above techniques, as the
forecast will be adjusted to these erroneous values. The D1/D2 MIX, the NBH method and smoothing (except for the first time
step) are less sensitive to observational errors than the other methods. Furthermore, the training data has to be representative
for the conditions at the site, including both seasonality and different types of synoptic and mesoscale situations. As suggested
by Fig. 6d, at least six months of training data is needed for the RF method, but preferably one to two years or more. It is
likely that the amount of training data needed for the other methods is similar. However, some methods (such as the smoothing
method), do not require any training data at all.

Even though the methods were only tested for one site, we believe that the main findings in the study are not site specific and
will generalize to other parts of the Baltic Sea and possibly also to other (offshore) areas. Even though LiDAR observations
are scarce in the Baltic Sea, the post-processing methods might as well be trained using wind speed data from e.g. a nacelle
mounted anemometer on an offshore wind turbine. More data in the test period would open up for a more detailed analysis of
the performance of the post-processing methods, for example how the methods perform in different wind speed bins given the
LWT.

In order to better represent the operational use of the AROME forecast, the delay of the forecast could be included in
the analysis. In reality, the AROME forecast is available approximately at 03 UTC and thus forecasts could be evaluated for
04 UTC to 03 UTC the following day, using the same forecast lengths as in our analysis. With observations available close to
real time it is then possible to assess the quality of the first three hours of the forecast and use this information in the subsequent
correction.

All methods can be implemented on a laptop and runtime is usually not an issue for small problems (one site, short forecasts).
For the RF, the runtime increases when adding more training features, increasing the number of trees or the training length or
when decreasing the MLS. The reason for the lower performance for MLS of 1 (as seen in Fig. 6c) is probably overfitting of
the data, using too many splits of the branches in the decision trees. More training features could be added to the RF to increase
performance, but it is clear from Table 2 that training features have to be relevant and not contain redundant information.
Examples of training features that provide additional information and could be tested are for example turbulent kinetic energy
and wave parameters.

The chain of adding training features one by one is implemented to provide as much insight as possible into the importance
of the individual training features. No more training features were added when the MAE skill score for the optimization period
started to decrease but this does not guarantee an optimal final set of features. For examples of other ways to arrive at an
optimal set of training features, we refer to Kursa and Rudnicki (2010); Huang et al. (2016); Cai et al. (2018); Shi et al. (2018).
To perfectly mimic how the RF could be optimized in a wind power forecasting company, the optimization period could be
selected from the historical data to be e.g. the same season, continuously adapting the training features and the settings in the
algorithm to the seasonal variations.

Although only the RF was tested here, there are many other possible ML methods that could be applied and the promising
result for the RF method should be interpreted as yet another indicator of the great potential benefit of applying ML methods
for post-processing.





As the wind field offshore is rather homogeneous, differences in wind speed at a specific time between grid points that are
close to each other is small, which explains the small changes to the forecast when applying the NBH method (Fig. 8). Onshore,
where spatial variation is higher, or if a model with a lower horizontal resolution is used, the method might still be important
to consider (Molinder et al., 2018). The NBH method smooths the forecast and just as for temporal smoothing it decreases the
risk of double penalty e.g. in cases of wind ramps in connection with a passing front zone or squall line. The increased skill
resulting from spatial smoothing of a NWP model is well studied, e.g. by Mass et al. (2002), and also temporal smoothing has
been shown to be beneficial for offshore wind power production before (Gilbert et al., 2020).

The reduced risk for double penalty is probably also the reason why the smoothing method performs best when the air flow
is from N or NE (Fig. 11), since winds from those directions typically are gusty due to the often unstable conditions related to
advection of cooler air. However, comparing different stabilities (Fig. 10) the smoothing method has similar performance in all
categories. It should also be mentioned that all post-processing methods that apply averaging (that is, all methods except the
LR) reduce the variability in the data and thus increase the risk of underestimating the occurrence of extreme conditions.

Comparing diurnal averages of wind power production data instead of hourly production, the correlation coefficient increases
to 0.97 and the CRMSE decreases to 300 kWh for the deterministic 0–23 h forecast. Using the post-processing methods based
on hourly data, only the NBH and smoothing methods result in positive MAE skill scores (0.007 and 0.005 respectively).
However, the performance of most methods could probably be improved if diurnal data instead of hourly data was used for
training.

The major improvement for the smoothing method is due to the interpolation of the observations at 23 UTC, as real-time
observations and short interpolations is the best way to increase forecast improvement for short forecasts. To develop the
smoothing function further, a running average with a longer block length duration could be used and time steps within the
block could be given different weights.

Applying smoothing directly to the power production forecast (instead of smoothing the wind speed forecast first and then
calculating the production) does not result in a higher MAE skill score for the test period, even though the correlation coefficient
and the CRMSE both slightly improve, see Fig. 13. Due to the shape of the power curve, error propagation in the wind speed
forecast is non-linear and using the smoothing method based on the forecasted power is better for wind speeds around cut-in
and the rated wind speed. In the ranges 2.5–5.0 m s$^{-1}$ and 15.0–17.5 m s$^{-1}$, the MAE skill scores for the smoothed method
based on power production are 0.24 and 0.10 respectively, to be compared with the results for the smoothing method based
on wind speed in Fig. 9. For the other wind speed bins, smoothing the wind speeds in the forecast is more beneficial than
smoothing the forecasted power production.

The synoptic classification using LWTs is beneficial compared to other synoptic classification methods as it simplifies the
meteorological reasoning and understanding of the forecast behaviour for different classes. One drawback with LWTs is,
however, that it is less strict from a mathematical perspective than for example principal component analysis (PCA) (Huth
et al., 2008). Also, the JC method has a tendency to classify the synoptic situation as anticyclonic or cyclonic too often and
hybrid classes and pure directional flow are less common. To get a sufficient amount of data in each class, we used the reduced





set of 11 LWTs instead of the original 27 classes (Demuzere et al., 2009). As a consequence of this, the variability of the pressure field within a class is quite large.

The D1/D2 MIX combines the D1 and D2 forecasts using a mix of approximately 60% of the data from the new forecast and 40% from the old. Only forecasts issued at 00 UTC have been used in this study but the method is general and directly applicable to all earlier forecasts that overlap in lead times.

    It is well known that NWP models struggle with resolving strongly stable stratification (Holtslag et al., 2013; Sandu et al., 2013). This can also be seen in Fig. 10b where this stability class has the highest CRMSE. Improving data assimilation and

schemes for turbulent mixing under these conditions are key to improve the forecasts (Reen and Stauffer, 2010; Wilczak et al., 2015) and would be beneficial for the representation of e.g. low-level jets and the extent of turbine wakes. It is a promising result that some of the post-processing methods tested, primarily smoothing and RF, managed to improve the forecasts under these conditions.

    The result in Fig. 4a was expected, as Utö is a pure offshore location in the AROME model (Fig. 1c) and stable (unstable)

stratification is known to be dominant in spring and summer (fall and winter) (Svensson et al., 2016). However, in contrast to these results, using tower data from Utö and calculating the stability for the 2–50 m layer, the stability varies with a pattern typical for boundary layers over land with a clear diurnal cycle during the summer months (stable during night and unstable during day) and mostly stable conditions during winter (results not shown). Even though the horizontal resolution in the model is high, many islands in the Finnish archipelago (such as Utö) are not resolved by the model as illustrated by the land/sea-mask

presented in Fig. 1b, and the issue with the stability is most probably a consequence of this.

    As the RF and smoothing are the methods with the best general performance (Fig. 7) it might be interesting to combine the two methods to see if this would further improve the forecast. This can be performed in two ways: either the RF is run as before and the resulting forecast is smoothed, or the training features are smoothed before the RF is applied. Both these methods were tested and the result is presented in Fig. 13. For the RF only the wind speed training features for D1 and D2 for the four grid

points (see Table 1) were smoothed. The RF was optimized in the same manner as described in Sect. 3.2.6.

    Figure 13 shows that none of the two setups managed to perform better than the smoothed forecast presented before. Still, smoothing the RF gave an improvement over the original RF method. It can also be seen that smoothing the training features for the RF improved the performance, but not as much as when applying smoothing afterwards. This is probably due to the fact that smoothing decreased the variability in the training data too much. The features selected in the optimization procedure

turned out to be the same as for the RF "wind speed to wind speed" setup (Table 2) with the exception of the final feature which was $Z$ for D2 instead of persistence. Since influence from the 23 UTC observation was embedded in the smoothed training features, it is reasonable that the persistence forecast was of secondary importance for this RF setup.

    Assuming that we would have known in advance that the RF was the best selection in case of an LLJ (Fig. 12), that the D1/D2 MIX excelled during weak synoptic flow (U) (Fig. 11) and that smoothing was the best method in general (Fig. 7),

we could have combined these methods accordingly. The result is presented in Fig. 13. Since the combination of methods is based on how well the methods performed for data in the test period, and was then evaluated for the same period, we know that this will improve the MAE skill score but not by how much. To properly investigate the effect of the combined forecast an

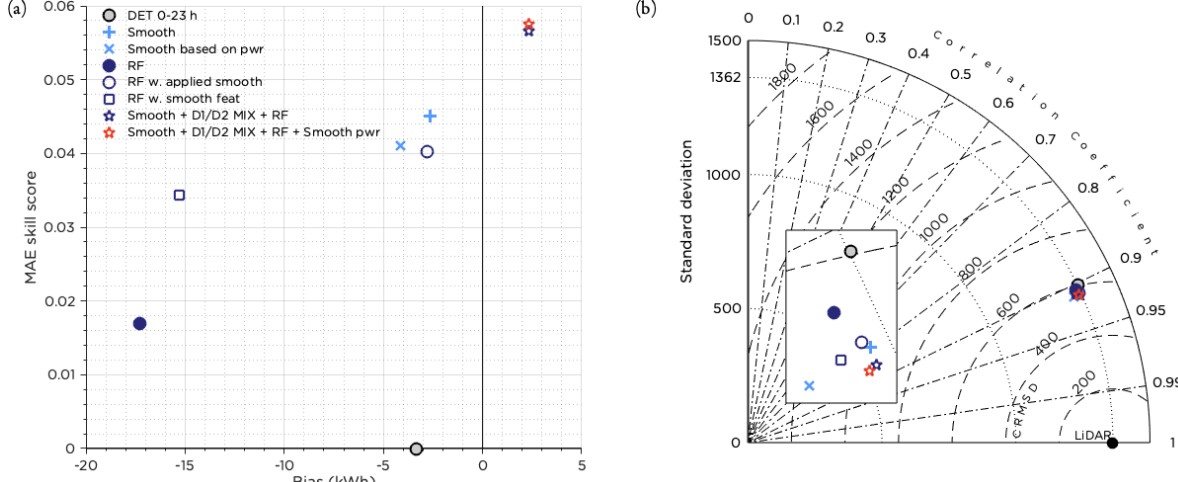

**Figure 13.** Performance of the RF with applied smoothing and the RF with smoothed features. Also the performance of the combined forecast using D1/D2 MIX (during weak synoptic forcing), RF (during LLJs) and smoothing otherwise is plotted. The result of adjusting the combined forecast using smoothing based on power production for wind speeds around cut-in and the rated wind speed is also plotted. In (a) the MAE skill score and bias is presented, in (b) the Taylor diagram. To simplify comparisons, the original deterministic forecast from AROME and the results for the smoothing and RF methods presented earlier are included. The inset in (b) shows an enlarged portions of the figure to more clearly show the differences between the methods.

independent test period is needed. Thus, the result in Fig. 13 can only be seen as an indicator of to what degree this combined method could have improved the forecast. The change in the MAE skill score is noticeable. Using the smoothing method based
on power production for wind speeds forecasted in the ranges 2.5–5.0 m s$^{-1}$ and 15.0–17.5 m s$^{-1}$ resulted in an additional minor improvement of the MAE skill score. For comparison with the results in Fig. 8, this combined forecast resulted in an EMD value of 331 kWh compared to a perfect forecast and was superior to the AROME D1 forecast 31% of the time. A more detailed investigation of the wind speed ranges where the smoothing method based on power production is better than the original smoothing and adjustment of the wind speed ranges when it is applied would likely result in further improvement of
the final forecast.

## 6  Conclusions

Six commonly used post-processing methods were applied to the 0–23 h AROME forecasts for wind speed at an offshore location in the Baltic Sea and evaluated in terms of performance in forecasting power production during one year.

Applying smoothing to the forecast or using a RF algorithm were the most promising methods to improve the forecast,
with the best performance in MAE skill score, highest correlation and lowest CRMSE. The smoothing method performed slightly better than the RF and had a lower bias. Combining the two techniques by smoothing the RF forecast or by smoothing





the training features before applying the RF algorithm improved the performance of the RF, but did not surpass the original smoothing method.

Even though the smoothing and RF methods improved the forecast for approximately 30% of the time, almost equally often
the methods deteriorated the forecast. For 40–50% of the time, the changes compared to the original forecast are small and in terms of forecast superiority and EMD the performance is only slightly better than what could be achieved by chance. However, the MAE skill score for the forecast with added noise is lower than for the smoothing method and RF.

The major improvement for the smoothing method is for the first time step of the forecast as interpolation of the 23 UTC observation was used. Both the smoothing and the RF worked well for intermediate wind speeds but for the RF both lower and
higher wind speeds were problematic. Smoothing improved the forecast in all stability classes while the RF mainly improved the forecast in stable and neutral stratification. Similarly, smoothing increased the performance for all synoptic situations while the RF struggled primarily with easterly winds, probably because the amount of training data for these situations was inadequate. RF improved the forecast substantially when a LLJ was forecasted, but gave almost no improvement otherwise.

Among the other methods tested, the D1/D2 MIX using a combination of the new forecast and the forecast from the previous
day, resulted in improvement under some conditions, but mainly when the synoptic flow was weak and/or neutral stratification. The best MAE skill score was achieved using smoothing and switching to RF in case of a LLJ and to D1/D2 MIX during weak synoptic forcing (and no LLJ). Further, if smoothing applied to the forecasted power production (instead of the wind speed) was used in the combined method for wind speeds in the ranges 2.5–5.0 m s$^{-1}$ and 15.0–17.5 m s$^{-1}$, an additional small improvement of the MAE skill score was achieved.

As the forecasted wind speeds for neighbouring grid points offshore are similar in a high-resolution NWP model, combining nearby grid points in a NBH method gave only minor changes compared to using the forecast from the closest grid point. Applying LR to improve the forecast was in general not a successful method.

The different post-processing methods applied in this study are all general and can be applied to any NWP model, any parameter and any forecast length offshore as well as onshore. For further studies, we suggest comparing state-of-the-art
ML methods in combination with nudging techniques to include real-time observations in the forecasts. Different methods to smooth the forecast or the training features for ML should be investigated. Also, an overview of different methods to improve probabilistic forecasts for offshore wind energy in the Baltic Sea would be of interest to the community.

**List of acronyms and abbreviations**

**AROME**   the forecast model used

**CRMSE**   Centered Root Mean Square Error

**D1/D2**   0–23 h and 24–47 h forecasts

**EMD**   Earth mover's distance

**JC**   Jenkinson and Collison



**LLJ** Low-level jet

**LiDAR** Light Detection And Ranging

**LR** Linear Regression

**LWT** Lamb Weather Type

**NBH** Neighbourhood

**NWP** Numerical Weather Prediction

**MAE** Mean Absolute Error

**ML** Machine Learning

**MLS** Minimum Leaf Size

**PCHIP** Piece-wise Cubic Hermite Interpolating Polynomial

**RF** Random Forest

**Appendix A**

Using only the instantaneous sea level pressure from 16 grid points, Jenkinson and Collison (1977) constructed an objective method to calculate the LWT valid for a focus area (Fig. 1a). In total there are 27 different LWTs (Lamb, 1972): anticyclonic (A), unclassified/weak flow (U), eight types of directional flow (N/NE/E/SE/S/SW/W/NW) and 16 types of hybrid flow (anticyclonic or cyclonic with a component of directional flow). The 27 classes can be reduced to 11 by treating all the hybrid flow

classes according to their directional component, following Demuzere et al. (2009).

The first step in the procedure to obtain the LWTs is to calculate the sea level pressure deviations from 1000 hPa at the grid points (Jenkinson and Collison, 1977). The zonal (westerly) and meridional (southerly) flow can then be calculated as

$$W = \frac{1}{2}\left(\mathbf{12}+\mathbf{13}\right) - \frac{1}{2}\left(\mathbf{4}+\mathbf{5}\right) \tag{A1}$$

and

$$S = \frac{1}{\cos\psi}\left(\frac{1}{4}\left(\mathbf{5}+2\cdot\mathbf{9}+\mathbf{13}\right) - \frac{1}{4}\left(\mathbf{4}+2\cdot\mathbf{8}+\mathbf{12}\right)\right) \tag{A2}$$

respectively. The bold numbers in the formulas indicate the sea level pressure deviations (in hPa) at their respective grid point (see Fig. 1a), following the notation in Jenkinson and Collison (1977). The variable $\psi$ is the latitude of the center line for the focus area where the LWT classification is valid. In this study we used $\psi = 63°$, resulting in the focus area marked in Fig. 1a. The reason for Utö not being in the center of the focus area is due to the domain size of AROME.





The resultant wind speed (in geostrophic units) is calculated as

$$ws = \sqrt{W^2 + S^2} \tag{A3}$$

and the wind direction is given by

$$wd = \tan^{-1}(W/S) \tag{A4}$$

with an addition of 180° if $W$ is positive (Jones et al., 2012). Using an eight-point compass rose (45° per sector) the wind
direction is classified.

    The westerly and southerly components of the shear vorticity (in geostrophic units) are

$$ZW = \frac{\sin\psi}{\sin(\psi-5)}\left(\frac{1}{2}(\mathbf{15}+\mathbf{16}) - \frac{1}{2}(\mathbf{8}+\mathbf{9})\right) - \frac{\sin\psi}{\sin(\psi+5)}\left(\frac{1}{2}(\mathbf{8}+\mathbf{9}) - \frac{1}{2}(\mathbf{1}+\mathbf{2})\right) \tag{A5}$$

and

$$ZS = \frac{1}{2\cos^2\psi}\left(\frac{1}{4}(\mathbf{6}+2\cdot\mathbf{10}+\mathbf{14}) - \frac{1}{4}(\mathbf{5}+2\cdot\mathbf{9}+\mathbf{13}) - \frac{1}{4}(\mathbf{4}+2\cdot\mathbf{8}+\mathbf{12}) + \frac{1}{4}(\mathbf{3}+2\cdot\mathbf{7}+\mathbf{11})\right) \tag{A6}$$

respectively and the total shear vorticity for the focus area can then be calculated as

$$Z = ZW + ZS \tag{A7}$$

    Based on these variables, the synoptic situation can be uniquely classified as one of the 27 LWTs following this scheme of
conditions (Jones et al., 2012):

1. If $ws < 6$ and $|Z| < 6$: Weak flow (unclassified), LWT is U

2. Else, if $|Z| \leq ws$: Directional flow, LWT is N/NE/E/SE/S/SW/W/NW

3. Else, if $|Z| > ws$ and $|Z| < 2 \cdot ws$: Hybrid flow that is either cyclonic (if $Z > 0$, LWT is e.g. CSE) or anticyclonic (if $Z < 0$, LWT is e.g. ASE)

4. Else, if $|Z| \geq 2 \cdot ws$ the LWT is either cyclonic (C) if $Z > 0$ or anticyclonic (A) if $Z < 0$.

*Author contributions.* The project was conceptualized and administrated by CH, ES, HK and SI. Funding acquisition: ES, HK, SI. The
methodology, programming, validation, formal analysis and visualization was performed by CH who also wrote the original draft. CH was
supervised by ES, HK and SI. Data curation: VV. All authors participated in reviewing and editing the manuscript.

*Competing interests.* The authors declare no conflict of interest. The funders had no role in the design of the study; in the collection, analyses,
or interpretation of data; in the writing of the manuscript, or in the decision to publish the results.



*Acknowledgements.* This research was funded by the Energimyndigheten (Swedish Energy Agency) VindEl program, Grant Number 47054-1.
The work forms part of the Swedish strategic research program StandUp for Wind. The AROME data used in the study was provided by The
Norwegian Meteorological Institute.



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
