# Peer review of "The smoother the better?"

_Wind Energy Science, 2021_

## Author Response (AR1)

July 21, 2021

*Wind Energy Science*

Dear Referees,

Thank you both for your positive feedback on our manuscript!

We have reviewed the manuscript according to your comments and made the following changes:

Referee #1

**Similarly testing at other locations is found as potential interesting future work. In any case, this paper may serve as a robust guideline for future tests, for instance with other NWP models.**

We stated more clearly in the Discussion that, for future work, we suggest similar tests to be performed at other (offshore and onshore) locations as well as based on other NWP models.

Lines 392-394 in the revised manuscript: Building on this study, we suggest for future work to compare the performance of the different post-processing methods for other offshore (and potentially also onshore) locations as well as testing other NWP models and using a longer time series of observations and forecasts.

**Although this is a research paper, I wonder how feasible it would be to implement this methodology for operative forecast purposes.**

The post-processing methods are applied to mimic how they could be used in operational forecasting. Thus, we believe that the results hold for forecasting purposes. However, we agree that real-time testing of the methods in operational mode would be of interest. We added a comment on this in the Discussion.

Lines 394-396 in the revised manuscript: Although the post-processing methods have been applied to mimic how they could be used in operational forecasting, a real-time test of the methods in fully operational mode would be of interest.

**Why are the months of the year grouped this way?**

The reason for using February 1st 2018 – January 31st 2020 is simply due to the fact that the AROME model used was updated in February 2020, and using data from different model cycles would not be comparable. It is possible to switch to e.g. January 1st 2018 – December 31st 2019, but (as is indicated by Fig. 2) LiDAR data quality was lower in beginning of 2018 and switching to this time period would have decreased the amount of training data. Thus, we decided to use as recent data as possible. We also wanted to keep the data to two full year cycles to minimize seasonal bias in the analysis. There is a comment on the AROME update in the Discussion, but we also added a footnote in the Materials section 2.1 to clarify the reasoning behind the choice of time period.

Footnote added in Section 2.1 in the revised manuscript: There was a major update of the AROME model (Sect. 2.2) in February 2020, thus the time period analyzed in this study was restricted to February 1st 2018 to January 31st 2020 in order to keep as recent data as possible. Two full year cycles were studied to minimize seasonal bias in the analysis

**Regarding the wind direction analysis, it is stated that the RF method didn't improve the forecast for easterly and northwesterly flows. The authors apparently justify this fact based on the lower amount of**

**data from these non-predominant directions. Is it then expected an improvement using a larger training period?**

This is an interesting question. In the wind direction analysis, Fig. 11, we see that the RF method did not improve the forecast for wind directions NE-SE (Fig. 11c) and also that there is an underestimation in the power production forecast for these wind directions (Fig. 11a). The easterly wind directions are the least frequent directions (Fig. 11c) and we believe that a longer training period would be beneficial for the RF method to build upon. It might also be the case that these wind directions are intrinsically more difficult to forecast, and also in this case a longer time series is crucial for the RF to find patterns in the data.

**Lines 386-387. Please justify this statement.**

References were added to justify the statement on lines 386-387 in the revised manuscript (Foley et al. 2012, Vannitsem et al. 2020)

**Maybe it is worth saying what AROME (or HARMONIE-AROME) states for in the acronyms list, instead of just mentioning it is the model used.**

The acronym AROME (Applications of Research to Operations at Mesoscale) is now explained both in the list and in the text (lines 98-99 in the revised manuscript)

**Would this analysis significantly change if data from several years ago were considered?**

This is an interesting remark. As the year-to-year variability in synoptic conditions is large in the Baltic Sea area, a longer time series of observations and forecasts (multiple decades) would be crucial to be able to fully conclude how much the results are dependent on selected individual years. Re-forecasts can be created, but longer time series of wind speed observations at hub height are not available for past conditions on that time scale. One way to explore the representativeness of individual years would be compare deviations to climatological conditions. A thorough analysis of how the selected individual years used in this study differ from normal conditions and how this might affect the performance of the different post-processing methods tested is beyond the scope of this manuscript. However, splitting the analysis into different wind speed bins, stabilities and synoptic weather types give some insight in the variability. Repeating the methodology for more sites, using other NWP models, testing different years would all be possible ways to contribute to estimate the uncertainty in the results. We added a comment regarding this in the Discussion. Thank you for this interesting question!

Lines 391-392 in the revised manuscript: Also, a detailed analysis of the representativeness of the years analyzed in this study would contribute to assessing the uncertainty in the results.

Referee #2

**Line 115: I might have missed it, but please spell out PCHIP at first use**

The acronym PCHIP (piece-wise cubic Hermite interpolating polynomial) is explained on line 87. Thus, we did not change anything on line 115.

**Table 1: Please indicate in the caption what the differences between the two columns are.**

We have rearranged this Table and added headers to clarify the difference between the columns (training features "Based on AROME D1", "Based on AROME D2", "Based on LiDAR" or "Based on index"). We also changed the text in the caption to the following:

Table 1. All training features available for the RF algorithm. In the first column, training features based on the most recent forecast (D1) are listed, and in the second column, training features based on the forecast from the day before (D2) are listed (see Sect. 3.2.1 for details). The indices denote the grid points as marked in Fig. 1c. The persistence forecast for wind speed, listed in the third column, was generated using the LiDAR observations at 23 UTC as described in Sect. 3.2.2. In the fourth column, training features based on index are listed. All forecasts for wind speed, wind direction and temperature are valid for hub height (90 m).

**Line 360: no ( ) for Hallgren et al., 2020.**

We have now fixed this.

References

Foley, A. M., Leahy, P. G., Marvuglia, A., and McKeogh, E. J.: Current methods and advances in forecasting of wind power generation, Renewable Energy, 37, 1–8, https://doi.org/10.1016/j.renene.2011.05.033, 2012.

Vannitsem, S., Bremnes, J. B., Demaeyer, J., Evans, G. R., Flowerdew, J., Hemri, S., Lerch, S., Roberts, N., Theis, S., Atencia, A., et al.: Statistical postprocessing for weather forecasts–review, challenges and avenues in a big data world, Bulletin of the American Meteorological Society, pp. 1–44, https://doi.org/10.1175/BAMS-D-19-0308.1, 2020

Best regards,

Christoffer Hallgren and co-authors

Christoffer Hallgren
Department of Earth Sciences
Uppsala University
Villavägen 16
752 36 Uppsala, Sweden
+460705863807
christoffer.hallgren@geo.uu.se